# Development of an Online Mind–Body Physical Activity Intervention for Young Adults during COVID-19: A Pilot Study

**DOI:** 10.3390/ijerph20054562

**Published:** 2023-03-04

**Authors:** Ildiko Strehli, Ryan D. Burns, Yang Bai, Donna H. Ziegenfuss, Martin E. Block, Timothy A. Brusseau

**Affiliations:** 1Department of Health and Kinesiology, University of Utah, Salt Lake City, UT 84112, USA; 2Marriot Library, University of Utah, Salt Lake City, UT 84112, USA; 3Department of Kinesiology, University of Virginia, Charlottesville, VA 22903, USA

**Keywords:** affect, exercise, mental health, yoga, young adult

## Abstract

The purpose of this study was to examine the association between the implementation of an online mind–body physical activity (MBPA) intervention and physical activity (PA), stress, and well-being in young adults during COVID-19. The participants were a sample of college students (*N* = 21; 81% female). The MBPA intervention was organized in four online modules that were administered asynchronously for 8 weeks with three separate 10 min sessions per week. The intervention components consisted of traditional deep breathing, diaphragm mindful breathing, yoga poses, and walking meditation. Objective PA behaviors were assessed using wrist-worn ActiGraph accelerometers, and stress and well-being data were collected using validated self-report instruments. A 2 (sex) × 3 (time) doubly multivariate analysis of variance test with a univariate follow-up showed that the % of wear time in light (LPA) and moderate-to-vigorous physical activity (MVPA) was significantly higher at the end of the intervention compared to baseline (LPA mean difference = 11.3%, *p* = 0.003, *d* = 0.70; MVPA mean difference = 2.9%, *p* < 0.001, *d* = 0.56). No significant differences were observed for perceived stress and well-being, and there was no moderating effect of sex. The MBPA intervention showed promise, as it was associated with higher PA in young adults during COVID-19. No improvements were observed for stress and well-being. These results warrant further testing of the intervention’s effectiveness using larger samples.

## 1. Introduction

The SARS-CoV-2 pandemic (COVID-19) affected all sectors of society at the beginning of 2020, and many institutes of higher education adopted distance learning as a strategy to mitigate exposure to the virus [1,2]. As a result, students changed their daily movement behaviors, such as lowering their physical activity (PA) [3]. Regular PA is essential for enhancing well-being, and it positively correlates with stress management; however, many college students are inactive and thus have an increased risk of developing metabolic disorders and mental health issues [4,5,6]. Although a physically active lifestyle is essential for healthy growth and to prevent chronic conditions as students age, PA behaviors decrease from high school through the college years while individuals undergo a social, academic, and environmental transition [7,8]. Declines in PA behaviors might have been exacerbated by the pandemic and thus required an effective multicomponent intervention [9,10].

Stress among students is exceedingly prevalent in higher education, and the transition to adulthood is a sensitive period, as the central nervous and musculoskeletal systems are still developing into the third decade of life [11,12]. The ability to sense and respond to stress-related bodily changes is of paramount importance [13]. More than 64% of college students reported personal health problems, 74.8% experienced difficulties related to their loved ones, and a significant association exists between stress and mental disorders during this time of life [14]. Increasing stress levels interferes with many aspects of health, including mental health, which can be mitigated by a higher sense of well-being induced by regular PA [4,7,10,12].

Mental health and engagement in health behaviors, such as PA, have been shown to have a reciprocal relationship with each other [4,5,6,7,10,11,12,13,14,15]. Improvements in mental health can improve the motivation to engage in PA, and certain types of PA can improve mental health. Developing a positive cycle between engagement in PA and mental health can yield additive benefits within the young adult population. Mind–body physical activity (MBPA) may provide a way to concurrently improve both PA and mental health [11,15]. MBPA interventions are a type of physical activity behavioral programming that involves mindful movements via a combination of yoga, tai chi, qigong, mindful walking, and meditation. Mindfulness is generally referred to as non-judgmental present-moment awareness and is a desired mental quality [16]. Since physical activity is any movement that raises energy expenditure above resting levels, we are defining movements that utilize mindfulness, including yoga, tai chi, qigong, mindful walking, and walking meditation, as “MBPA”. MBPA can concurrently improve the total physical activity volume, including light and moderate physical activity, by improving the motivation to engage in physical activity during and after a mindfulness session [17,18]. The light and moderate movements that are involved in MBPA have been shown to improve mental health in young adults [19,20]. Indeed, a single bout of 10 min of walking, meditating, or a combination of walking and meditating can result in anxiolytic benefits among healthy students [21]. These effects are also seen in younger individuals in secondary educational settings, as previous work has shown that integrating mindfulness training in school health education can improve PA [22]. Indeed, ancient holistic MBPA practices are becoming powerful complementary lifestyle routines to manage physical and mental challenges, and there is some evidence to suggest that these interventions can improve PA behaviors [23,24,25]. However, there is limited research on brief MBPA interventions for students in higher education. Examining the preliminary efficacy of an MBPA intervention may have merit before designing and executing larger, more rigorous studies. Additionally, although the use of wrist-worn accelerometers to assess PA in research studies comes with several challenges, including complex device placement and data processing decisions [26], the use of accelerometers may improve the validity of findings from an MBPA intervention. Therefore, the purpose of this pilot study was to test the association between an online and asynchronous 8-week MBPA intervention and objective PA, perceived stress, and well-being in a sample of college students during COVID-19. We hypothesized that the primary outcome of PA would slightly improve after the 8-week MBPA intervention and that the secondary outcomes of perceived stress and well-being would moderately improve.

## 2. Materials and Methods

### 2.1. Study Protocol

This study was conducted during the 2021 spring semester from February to April. All participants were enrolled in college classes for credits towards graduation from a university within the western United States (US). Each student wore wrist-worn accelerometers at 3 assessment timepoints and completed self-report surveys at 6 assessment timepoints. Accelerometer data were collected at only 3 timepoints to facilitate study feasibility and to study the acceptability among the participants. During the pre-intervention phase (timepoints 1–3), self-report surveys were completed once per week for 3 separate weeks before the start of module 1. Postintervention timepoints for perceived stress and well-being (timepoints 4–6) were collected during module 2 (the end of week 3), during module 3 (the end of week 5), and during module 4 (the end of week 7). Accelerometers were worn and returned before starting module 1, during module 2 (the end of week 3), and at the end of module 4 (the end of week 8). Participants were given 2 weeks to complete each module. A study flow diagram is presented in Figure 1. All self-report assessments were administered using Canvas, and Zoom (San Jose, CA, USA), a communication video meeting platform, provided remote real-time interaction, as needed. Participants received an identification number, which was used for all research documents. To protect privacy, all personal information that was shared during the study was de-identified.

The study was conducted in accordance with the Declaration of Helsinki and was approved by the Institutional Review Board of the University of Utah (#IRB_00139334; 20 December 2020). The IRB application included online informed consent procedures, assessment items, and study protocols. Written consent was obtained from all participants before they were enrolled in the MBPA intervention. This research project used a non-randomized single-arm (one-group) pretest–post-test design for PA and an interrupted time series design for perceived stress and well-being. The study was not registered in a World Health Organization accredited trial registry. All participants received USD 100 in compensation for their time and effort following the completion of the intervention if they successfully provided data at each timepoint and returned the accelerometers to the researchers. Both quantitative and qualitative data were collected. The qualitative results were published elsewhere [27].

### 2.2. Physical Activity Assessment

PA was objectively monitored for 7 days using wrist-worn ActiGraph GT9X accelerometers. In comparative research studies, wrist-worn accelerometers have higher compliance rates compared to other placement sites [28,29,30,31]. Accelerometers provide valid, reliable, noninvasive, and objective measures of PA and have tracked frequency, intensity, duration, and useful in field settings [31,32]. The ActiGraph GT9X accelerometer captured movement in 3 axes, and PA was assessed at a rate of 100 Hz and within 60 s epochs. Light PA values were determined as 100 to 1951 counts/minute, and moderate-to-vigorous PA values were determined as 1952 counts/minute and higher [33,34]. Participants wore an ActiGraph GT9X tri-axial accelerometer at baseline, at the intervention midpoint, and at the end of the intervention. Participants were instructed to wear the devices for 24 h/day for 7 consecutive days at each timepoint, except for bathing, showering, and/or swimming. Valid wear time consisted of wearing the accelerometer for at least 3 weekdays and 1 weekend day for at least 10 h per day [35]. All participants in the analyzed sample provided valid accelerometer wear time data. ActiLife 6 software was used to initialize the accelerometers before distribution. PA data were analyzed as the % of wear time within each intensity to adjust for differences in device wear time across participants. Contextual data regarding specific PA behaviors that were engaged in outside of the wear time were not collected. Specific intervention activities are outlined in the Appendix A.

### 2.3. Assessment of Perceived Stress

The 4-item Perceived Stress Scale (PSS-4) was used to measure perceived stress [36,37]. The construct validity of the PSS-4 instrument has been established in prior work [36,37]. The PSS-4 responses were measured on a 5-point Likert scale ranging from “never” (=0) to “very often” (=4) with items that asked about situations in a student’s life in the last month that were perceived as stressful, such as “ability to handle personal problems” and “ability to control the important things in life”. A lower score represented healthier stress levels. With a Cronbach’s alpha of 0.77, the internal consistency of the PSS-4 was determined to be acceptable [37].

### 2.4. Assessment of Well-Being

The 5-item WHO Well-Being Index (WHO-5) was used to measure well-being (WHO-5, World Health Organization 1998). The 6-point Likert response scale from “all of the time” (=5) to “at no time” (=0) of the five questions referred to feelings over the last two weeks, such as cheerful, calm, vigorous, and rested, and a higher score represented better well-being. The established WHO-5 is one of the most used well-being surveys globally and is a screening tool for depression with a specificity of 0.81 and a sensitivity of 0.86 [30]. The scale has demonstrated acceptable psychometric properties, such as internal consistency and construct validity [30,38].

### 2.5. Description of Intervention

The tailored MBPA intervention used the core components of (1) traditional deep breathing and diaphragm mindful breathing activities; (2) yoga poses (asanas), breathing activities (pranayama), and qigong movements; and (3) loving-kindness walking meditation with present-moment awareness. Additional information regarding the intervention content is provided in the Appendix A. A researcher created the online class based on best practices, which was delivered using an asynchronous model where intervention participants had access to the pre-designed MBPA activities at any time and completed the activities at their own pace following the four outlined modules [39]. The intervention was organized in 4 modules and was conducted asynchronously online for 8 weeks with 3 approximately 10 min sessions per week via Canvas. This MBPA intervention duration was in line with the literature of general PA interventions [40,41,42]. The researcher provided pre-recorded videos and voice audio of the MBPA intervention protocol, and participants were asked to practice it any time during the week. The brief, 10 min practice took place in an environment where students felt comfortable or when they sensed that the MBPA intervention may provide a stress-alleviating experience. The researcher was a certified instructor with over 35 years of teaching experience.

### 2.6. Statistical Analysis

Because this was a pilot study, no a priori power analysis was conducted, and the efficacy/effectiveness was not tested. However, a post hoc power analysis was conducted. The achieved statistical power was 84.2% for the LPA outcome, 92.1% for the MVPA outcome, 44.4% for perceived stress, and 15.5% for well-being. The data were screened for outliers using z-scores (using a +/−3.0 z cut-point). Histograms were used to verify the assumption of normality of the residuals. Descriptive statistics are reported as means and standard deviations for continuous data and as counts and percentages for categorical data. To justify the use of multivariate (multiple dependent variable) analyses, the correlations among the outcomes at baseline were examined using Pearson product-moment correlations. The primary analysis for the PA outcomes consisted of a 2 (sex) × 3 (time) doubly multivariate analysis of variance test. A follow-up consisted of mixed-design analysis of variance (ANOVA) tests to examine the trends across time for the % of accelerometer wear time in LPA and MVPA and the moderating influence of sex. Sex was a two-level between-subjects factor, and time was a three-level within-subjects factor. The analysis for the stress and well-being outcomes consisted of a 2 (sex) × 6 (time) doubly multivariate analysis of variance test using a univariate ANOVA follow-up with sex as a two-level factor and time as a six-level factor. Wilks’ lambda was used to determine the significance of both multivariate models. The main effect of time was of primary interest within all models. To assess the assumption of sphericity, Mauchly’s test was employed, with a Greenhouse–Geisser correction factor applied to the degrees of freedom if violated. The effect sizes of the main effects were examined using partial eta-squared. Pairwise comparison effect sizes were examined using Cohen’s delta (*d*) and were considered small if *d* = 0.20, medium if *d* = 0.50, and large if *d* = 0.80 [43]. The alpha level was set at *p* ≤ 0.05 but was adjusted using the Bonferroni correction for multiple comparisons. All analyses were conducted using IBM SPSS Statistics Version 27 (IBM, Armonk, NY, USA).

## 3. Results

### 3.1. Descriptive Statistics

Thirty participants were initially recruited using a convenience sampling methodology and completed the informed consent process. Twenty-six participants completed the pre-intervention surveys and received the accelerometer. However, five participants were lost to follow-up (21/26, or 80.8% retention) due to COVID-19-related issues. Participants spent a range of 2 to 48 h on the intervention’s Canvas webpage with a range of 19 to 44 video plays per module (see Section 2.6). Thus, the analyzed sample consisted of 21 participants (17 females and 4 males) who completed all assessments. The demographics for the analyzed sample are presented in Table 1. An inspection of histograms showed approximately normal distributions for each of the outcomes’ residuals. The descriptive statistics at each timepoint are provided for PA in Table 2 and for stress and well-being in Table 3.

### 3.2. Baseline Correlations

The Pearson product-moment correlations among the outcomes are presented in Table 4. There was a strong and positive bivariate correlation between the % wear time in LPA and MVPA at baseline (*p* < 0.001), a moderate and negative correlation between LPA and stress (*p* = 0.026), and a moderate and negative correlation between stress and well-being (*p* = 0.015). These significant bivariate correlations justified the use of multivariate analyses instead of separate univariate analyses. No other significant correlations between outcomes were observed at baseline.

### 3.3. Multivariate Models

The PA doubly MANOVA model was statistically significant for the main effect of time (Wilks’ lambda = 0.78, *p* = 0.008). The doubly MANOVA model for the main effect of time for stress and well-being was not significant (Wilks’ lambda = 0.95, *p* = 0.751). Additionally, the sex × time interaction was not significant within either the PA MANOVA model (Wilks’ lambda = 0.99, *p* = 0.977) or the stress and well-being MANOVA model (Wilks’ lambda = 0.95, *p* = 0.751). The ANOVA tables for each of the univariate follow-up analyses are reported in Table 5.

### 3.4. Univariate Models

Mixed-design ANOVA follow-up tests showed that there was a significant main effect of time on LPA (F(2,37) = 6.43, *p* = 0.004, partial eta-squared = 0.26), with higher LPA values postintervention compared to baseline (mean difference = 11.3%, *p* = 0.003, *d* = 0.70). Additionally, there was a significant main effect of time on MVPA (F(2,37) = 8.12, *p* = 0.001, partial eta-squared = 0.30), with higher MVPA scores postintervention compared to baseline (mean difference = 2.9%, *p* < 0.001, *d* = 0.56).

Mixed-design ANOVA follow-up tests showed that there were no significant main effects of time on perceived stress (F(2,37) = 0.88, *p* = 0.499, partial eta-squared = 0.04) and well-being (F(2,37) = 0.63, *p* = 0.678, partial eta-squared = 0.04).

## 4. Discussion

The purpose of this pilot study was to examine the association between an online and asynchronous MBPA intervention and objective PA behavior, perceived stress, and well-being among college students during COVID-19. The principal findings indicate statistically significant main effects for the time spent in LPA and MVPA, displaying greater postintervention scores compared to baseline. The magnitude of the effects was similar for both LPA and MVPA, possibly because the MBPA interventions incorporated both light- and moderate-intensity movements. However, the results also demonstrated no significant main effects of time on self-reported stress and well-being. Overall, the results suggest encouraging trends for the PA outcomes that were tested and indicate that further investigation is needed in a more extensive study with larger samples and a more rigorous design to make causal inferences, which was precluded in the current study, given the design and the lack of thorough fidelity and adherence data. Further discussions of our findings are provided below.

Prior research has found significant negative associations between increased PA and mental health problems, with the most robust effect sizes for frequency of activity [4]. Previous research on healthy behaviors among students in higher education revealed that PA levels significantly decreased while sedentary and sitting time increased during the COVID-19 pandemic worldwide [44]. Conversely, our results suggest that participation in the online MBPA intervention is associated with higher PA at both light and moderate levels. Previous research investigated MBPA interventions such as yoga, qigong, tai chi, and mindfulness walking activities within their frameworks, as they were developed in ancient cultures; we integrated these movements, as they are all generally considered light- and moderate-intensity PA [11,23]. Besides ancient movements such as yoga, qigong, and tai chi, walking activities were some of the core elements of our MBPA intervention. An early systematic review of PA levels conducted in eight countries observed that PA levels substantially decreased, such as an objective reduction of 67.7% fewer daily steps, during COVID-19 [44]. Specifically, young adults spent increased time using smartphones, sleeping, and sitting while decreasing time performing PA [35]. From the total of ten systematically reviewed studies, significant reductions in PA levels were found in nine investigations; in one study, students spent significantly more time both sitting and in PA [45]. Given that regular PA is a complex behavior to capture, using objective instruments to assess PA using accelerometers presents its challenges and values [11]. Furthermore, the majority of researchers applied self-administered questionnaires, while in one study, accelerometers were used to capture PA behavior during COVID-19 among 20 university students and found no significant increase in PA levels [45]. Although local confinements have altered and affected health behaviors and most studies reported significant decreases in PA levels globally, one study found substantial increases in MVPA and the total number of minutes of PA per week, which were self-reported by health sciences students [45,46].

Students perceive that frequent light and moderate PA could help maintain well-being [45]. Similarly, in the present study, participants were physically active three times weekly for 10 min while practicing the MBPA intervention. Concurrently, researchers found significant negative associations between PA intensity, duration, and frequency and mental health problems, with the most robust effect size for frequency, among 50,054 college students [4]. Moreover, researchers indicate that PA levels decrease from freshman to seniors and do not meet guideline recommendations [5]. Encouraging and providing the opportunity to frequently engage in brief MBPA could be an effective way to increase PA and cope with academic and interpersonal stress, thus enhancing overall well-being [47,48]. Our findings are encouraging because they indicate that the MBPA intervention has the potential to be associated with PA and may also have positive associations with physical and mental health. Similarly, a meta-analytical review found statistically significant and large pooled effects for an MBPA intervention’s effectiveness in lowering health-related physiological health markers, such as cortisol and heart rate [24,25].

Our findings revealed no statistically significant main effects for perceived stress and well-being; however, the results demonstrated non-significant changes during the intervention. These changes in a favorable direction could be attributed to the enjoyment of the MBPA activities and learning new skills. From the PA perspective, enjoyment and neural changes (i.e., neuroplasticity) occur when learning new skills, which are present across the lifespan [49,50]. The common goal of the MBPA intervention was to integrate mind, body, and spirit (i.e., vitality, lifeforce, and energy) while being physically active and aware of the breath, thus improving physical and mental health and well-being with stress alleviation, all in one package [11,18]. Light and moderate (i.e., walking) PA combined with purposeful attention to the breath, asanas, and loving kindness in a safe environment may create a positive physiological and psychological state and may influence stress-related health markers (i.e., cortisol, heart rate, and blood pressure) and neurological processes in favorable ways as a beneficial experience in young adults [51,52]. Additionally, researchers found significant negative associations between PA intensity, duration, and frequency and mental health problems, with the most robust effect size for frequency [4]. Similarly, longitudinal research among college students on loving kindness and PA revealed significant decreases in depression, stress, and anxiety, and higher PA levels indicated better moods on the same day and on following days [7,53]. Despite, these changes, no inferential conclusions can be made, given the lack of statistical significance.

First-year college students experience environmental, psychological, physiological, and academic changes; thus, it is vital to increase PA levels as an individual and institutional responsibility. Consequently, regular programs need to be incorporated to overcome the inevitable encounters of life and maintain overall well-being. Given the overall results and practical implications, we can state that MBPA applied in higher education is an essential lifestyle behavior to meet the needs of students and institutions in the COVID-19 era and beyond. Therefore, a brief multilevel and multicomponent 10 min MBPA is a viable, feasible, and effective solution for optimal well-being, as we discuss further. The MBPA program can be introduced to all incoming students and can create evenly distributed practice environments around campus and online to structure MBPA learning opportunities. To protect and promote students’ well-being, understanding their experience both on the intra- and interpersonal levels should be acknowledged and related to personal and academic success.

The results of this study should be interpreted within the context of its strengths and limitations. The strength of our research includes the use of a novel MBPA intervention that was delivered to a college student (young adult) sample during COVID-19. We also used an objective PA assessment to capture LPA and MVPA across the entire day. The study’s primary limitation was its small sample size and the fact that there was no active control group. This limits the internal validity of the results, as the relationships observed in the study are correlations and not causal effects. However, we would like to reiterate that this was a pilot study conducted during COVID-19; therefore, study rigor was attenuated, and more of a focus was placed on study feasibility and acceptability among the participants. The results are not generalizable to younger or older age groups. Moreover, response bias from the self-report instruments may have been present in the data. An additional limitation of this study was the lack of an examination of other moderating variables and mediating variables to determine potential differential patterns or mechanisms of effect. We did not collect contextual data for PA, such as the types of activities that the participants engaged in outside of the intervention. The MBPA needs further investigation because, as a multicomponent intervention, without component analyses, we cannot tell to what degree the online (synchronous vs. asynchronous) delivery or the distribution of ancient movements specifically influenced the outcomes.

## 5. Conclusions

In conclusion, the online MBPA intervention was associated with higher LPA and MVPA during the COVID-19 crisis as students progressed through the semester. Data on participants’ self-reported perceived stress and well-being suggested no significant changes throughout the intervention. Thus, the MBPA shows the potential to be a viable intervention to is associated with more PA behaviors; however, there is no evidence that this MBPA intervention is associated with lower stress or better well-being. Future research needs to further examine the MBPA intervention using objective measures, such as stress-related physiological health markers or salivary markers of inflammation, and employ more rigorous research designs and larger samples to improve the internal validity of the findings. MBPA was indeed found to be a credible approach to improve PA behaviors. A theoretical mechanism for this positive outcome is an improved overall state of mental health during the MBPA intervention, as mental health has previously been shown to be a predictor of PA behavior. However, because no quantitative improvements were observed for perceived stress and well-being, it is unclear if overall mental health or specific components of mental health were mediators of the effect between the intervention delivery and the increased levels of PA. Future research should examine potential mediating mechanisms of effect along with the possible influences of other behaviors such as sleep. Future research should also test the MBPA intervention using more rigorous experimental designs that are fully powered to detect the significance of all outcomes. Doing so will allow researchers to test for intervention effectiveness rather than association. Interventions such as MBPA have demonstrated meaningful associations with young adults’ PA during the COVID-19 pandemic.

## Figures and Tables

**Figure 1 ijerph-20-04562-f001:**
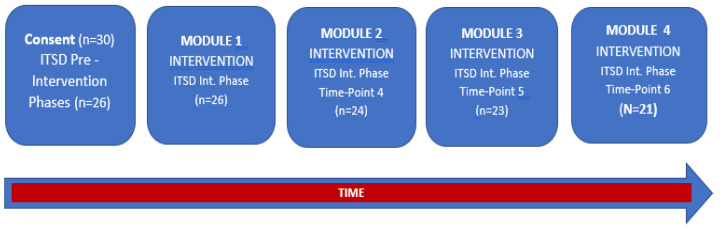
Study flow diagram. Note: ITSD stands for interrupted time series design; Int. stands for intervention. Participants were provided 2 weeks to complete each module once it was started on Canvas.

**Table 1 ijerph-20-04562-t001:** Demographic characteristics of the analyzed sample (*N* = 21).

Characteristic	Mean ± *SD* or *N* (%)
Age (years)	21.0 ± 2.2
Height (m)	1.7 ± 0.1
Weight (kg)	69.5 ± 17.5
Body Mass Index (kg/m^2^)	24.5 ± 4.5
Sex	
Male	4 (19%)
Female	17 (81%)
Race/Ethnicity	
White	17 (81%)
Hispanic or Latino	2 (9%)
Black or African American	1 (5%)
Asian or Pacific Islander	1 (5%)
University Status	
Freshman	5 (24%)
Sophomore	4 (19%)
Junior	6 (28%)
Senior	5 (24%)
Graduate	1 (5%)

Note: *SD* = standard deviation.

**Table 2 ijerph-20-04562-t002:** Descriptive statistics for objective physical activity at each timepoint.

Variable	Baseline (Mean ± *SD*)	Midpoint (Mean ± *SD*)	Postintervention (Mean ± *SD*)
Time in LPA (% of device wear time)	15.0 ± 14.8	**24.9 ± 16.3 ****	**26.3 ± 16.3 ****
Time in MVPA (% of device wear time)	3.9 ± 3.9	**7.3 ± 5.0 *****	**6.8 ± 5.1 *****

Note: *SD* = standard deviation; LPA = light physical activity; MVPA = moderate-to-vigorous physical activity; bold denotes statistical significance, ** *p* < 0.01, *** *p* < 0.001.

**Table 3 ijerph-20-04562-t003:** Descriptive statistics for stress and well-being at each timepoint.

Variable	Timepoint 1 (Mean ± SD)	Timepoint 2 (Mean ± SD)	Timepoint 3 (Mean ± SD)	Timepoint 4 (Mean ± SD)	Timepoint 5 (Mean ± SD)	Timepoint 6 (Mean ± SD)
Stress (PSS-4)	6.55 ± 2.61	6.71 ± 2.36	6.81 ± 2.67	5.81 ± 2.29	5.76 ± 3.17	6.00 ± 3.22
Well-Being (WHO-5)	14.05 ± 3.79	14.23 ± 3.76	13.95 ± 4.11	15.57 ± 3.82	15.66 ± 4.84	14.10 ± 5.69

Note: *SD* = standard deviation; PSS-4 = Perceived Stress Scale—higher values indicate more stress; WHO-5 = World Health Organization Well-Being Index—raw scores range from 0, representing the worst imaginable well-being, to 25, representing the best imaginable well-being.

**Table 4 ijerph-20-04562-t004:** Pearson product-moment correlations between outcomes at baseline.

	LPA	MVPA	Stress	Well-being
LPA	1			
MVPA	**0.86 *****	1		
Stress	**−0.52 ***	0.41	1	
Well-being	−0.44	-0.39	**−0.52 ***	1

Note: LPA = light physical activity; MVPA = moderate-to-vigorous physical activity; bold denotes statistical significance, * *p* < 0.05, *** *p* < 0.001.

**Table 5 ijerph-20-04562-t005:** Analysis of variance table for each of the targeted outcomes.

Outcome	Source	Sum of Squares	Degrees of Freedom	Mean Square	F-Statistic
Time in LPA (% of wear time)	Sex	346.1	1	346.1	0.51
	Time	741.1	2	379.8	**6.43 ****
	Sex × Time	13.5	2	6.7	0.12
Time in MVPA (% of wear time)	Sex	132.5.	1	132.5	0.53
	Time	86.7	2	43.4	**8.12 ****
	Sex × Time	1.3	2	0.6	0.12
Stress	Sex	5.96	1	5.96	0.23
	Time	18.68	5	3.73	0.88
	Sex × Time	12.20	5	2.44	0.72
Well-being	Sex	83.7	1	83.7	1.22
	Time	29	5	5.74	0.63
	Sex × Time	15	5	3.06	0.34

Note: LPA stands for light physical activity; MVPA stands for moderate-to-vigorous physical activity; bold denotes a statistically significant F-statistic, ** *p* < 0.01.

## Data Availability

The dataset of the current study is available from the authors upon request.

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
