# Peer review of "Development of an Online Mind–Body Physical Activity Intervention for Young Adults during COVID-19: A Pilot Study"

_ijerph, 2023, doi:10.3390/ijerph20054562_

Round 1

Reviewer 1 Report

This study examined the association between implementation of an online Mind-Body Physical Activity (MBPA) intervention with physical activity (PA), stress, and wellbeing in young adult, college-students during COVID-19. A major strength to this study is that the authors delivered the intervention remotely. This pilot-study demonstrated that MBPA can increase light and moderate-to-vigorous physical activity in college students. The manuscript is generally nicely written and clear. The reviewer provides some suggestions for revisions. 

Introduction

·      The introduction is very well written, clear, and funnels to the problem and hypothesis nicely. 

·      The reviewer does have a few minor grammatical suggestions:

o   Pg 2, ln 47: Add “exists” to “a significant association exists between life”

o   Pg 2, ln 59-60: You already defined MBPA in lines 53-54

o   Pg 32, ln 74: Place a comma after designing

Methods

·      Section 2.1. does not really address the recruitment of participants or your study participants inclusion/exclusion criteria. In fact, most of the information presented here (lns 89-95 and 97-99) should be in the results sections. Further, lns 86-88 and 95-97 can be placed in the statistical analysis section. Therefore, this section can be eliminated.

·      Figure 1: Please define abbreviations used in the figure. Further, it is not clear from the figure what the time points mean and how they align with the 8 week intervention. Please clarify when during the intervention each module was introduced and expected to be completed by. 

·      Please make more explicitly clear in the methods when testing was done in relation to the 8 week intervention. Was post-intervention done after the 8 weeks or in week 8? What weeks did the 6 timepoints for the stress and well-being measures collected?

·      Pg 2, ln 91: Do you mean it took place during the months of February through April?

·      Pg 3, ln 106: spell out United States on first use

·      Pg 5, lns 171-175: The reviewer does not feel it is necessary to site the examples in the methods. It is sufficient to cite references 40-42 after PA interventions in line 171.

·      Pg 5, ln 182-187: These examples do not add clarity for the reader and are not necessary to include. Further, you already provided the benefits of the MBPA in the intro, so it does not need to be repeated in the methods.

·      Please ensure that all of the methods are written in the past tense. Further, some of the sentences written in the methods (e.g., the brief, 10-minute practice should take place in an environment where students feel comfortable) sounds as though you are giving instructions and not outlining the methods of the study.

·      While it is a strength of the study that the authors collected objective measures of PA using accelerometers, did the authors collect information on the types of activities that individuals were engaging in (walking, yoga, tai chi) etc.? This information could be extremely valuable for designing future interventions as well as interpreting why there may not have been significant changes in perceived stress or well-being. 

Results

·      Pg 6, ln 219-220: The reviewer does not feel it is necessary or adds to interpretation of the data to include the histograms. It is adequate to state that data was normally distributed (or not). 

·      Pg 6, ln 220-222: Please rewrite. 

·      Pg 6, ln 222-224: This should be included in the statistical analysis section

·      Why did the authors only conduct baseline correlations and not complete correlations using the same variables post-intervention?

Author Response

Pg 2, ln 47: Add “exists” to “a significant association exists between life”                                           -Thank you. This has now been added (line 48).

Pg 2, ln 59-60: You already defined MBPA in lines 53-54                                                                    -Thank you. This redundant definition has now been removed (lines 63–64).

Pg 32, ln 74: Place a comma after designing.                                                                                             -Thank you. The comma has now been added to that sentence (line 76).

Section 2.1. does not really address the recruitment of participants or your study participants inclusion/exclusion criteria. In fact, most of the information presented here (lns 89-95 and 97-99) should be in the results sections. Further, lns 86-88 and 95-97 can be placed in the statistical analysis section. Therefore, this section can be eliminated.                                                                                            -Thank you. We have now eliminated the former section 2.1 and have renumbered the subsections within the Methods. We have transposed the content in the eliminated section to the Statistical Analysis and Results sections as the Reviewer suggested (lines 203–206; 283–290).

Figure 1: Please define abbreviations used in the figure. Further, it is not clear from the figure what the time points mean and how they align with the 8 week intervention. Please clarify when during the intervention each module was introduced and expected to be completed by.                                                                                                                                                      -Thank you. We now spell out abbreviations in Figure 1 in a note below the Figure (lines 135-136). Participants were provided 2 weeks to complete each Module once started on Canvas. This has now been indicated in the Note below Figure 1.

Please make more explicitly clear in the methods when testing was done in relation to the 8 week intervention. Was post-intervention done after the 8 weeks or in week 8? What weeks did the 6 timepoints for the stress and well-being measures collected?                                                                                                            -Thank you. This information has now been provided within the study protocol section in the Methods (new Section 2.1; lines 94–116).

Pg 2, ln 91: Do you mean it took place during the months of February through April?                                                                                                                                      -Yes, we now changed the wording in this sentence (lines 88–89).

Pg 3, ln 106: spell out United States on first use                                                                                      -Thank you. We now spell out United States within this sentence (line 90).

Pg 5, lns 171-175: The reviewer does not feel it is necessary to site the examples in the methods. It is sufficient to cite references 40-42 after PA interventions in line 171.                                                                                                                                   -Thank you. These sentences have been removed within the Methods section (line 192–198).

Pg 5, ln 182-187: These examples do not add clarity for the reader and are not necessary to include. Further, you already provided the benefits of the MBPA in the intro, so it does not need to be repeated in the methods.                                                                                                                   -Thank you, these sentences have been removed in the Methods section (line 192–198).

Please ensure that all of the methods are written in the past tense. Further, some of the sentences written in the methods (e.g., the brief, 10-minute practice should take place in an environment where students feel comfortable) sounds as though you are giving instructions and not outlining the methods of the study.                                                                                                                        -Thank you for the comment. We edited the Methods section to ensure that the writing was in past tense. The awkward sentences have been revised throughout.

While it is a strength of the study that the authors collected objective measures of PA using accelerometers, did the authors collect information on the types of activities that individuals were engaging in (walking, yoga, tai chi) etc.? This information could be extremely valuable for designing future interventions as well as interpreting why there may not have been significant changes in perceived stress or well-being.                                                                                           -We did not collect data on activities participated in outside of the intervention. We now indicate this in the Methods section and now mention this as a major limitation of the study (line 154–156; 471–472).

Pg 6, ln 219-220: The reviewer does not feel it is necessary or adds to interpretation of the data to include the histograms. It is adequate to state that data was normally distributed (or not).                                                                                                                                                           -Thank you. The histograms have now been removed from the manuscript.

Pg 6, ln 220-222: Please rewrite.                                                                                                             -Thank you. This sentence has been rewritten (line 291–292).

Pg 6, ln 222-224: This should be included in the statistical analysis section.                                       -Thank you. This power analysis content has been moved to the beginning of the Statistical Analysis section (line 203–206).

Why did the authors only conduct baseline correlations and not complete correlations using the same variables post-intervention?                                                                                                             -Thank you. We wanted to justify using multivariate analyses (MANOVA) instead of separate univariate analyses by showing that there was some correlation among specific outcomes at the baseline timepoint. We now indicate this in the Statistical Analysis and Results sections (lines 209–210; 343-344).

Reviewer 2 Report

1.          In the section of the introduction, I suggest the author could more information about the developing process or theoretical development of Mind-Body Physical Activities (MBPAs).

2.          In the section of the materials and methods, I suggest the author could provide more information about the details in figure 1.

3.          In the section of the assessments, I suggest the author should provide more information about the theoretical discourses, validity, and reliability of these measures of the physical activity assessment, perceived stress, and wellbeing.

4.          In the section of the conclusion, I suggest the author should provide theoretical reflections and practical suggestions for the development of this pilot study.

Author Response

In the section of the introduction, I suggest the author could more information about the developing process or theoretical development of Mind-Body Physical Activities (MBPAs).                                                                                                               -Thank you. This additional content has now been added to the Introduction section (lines 52–57).

In the section of the materials and methods, I suggest the author could provide more information about the details in figure 1.                                                                                                                   -Thank you. This additional information has now been provided within the study protocol section in the Methods (new Section 2.1; lines 94–116).

In the section of the assessments, I suggest the author should provide more information about the theoretical discourses, validity, and reliability of these measures of the physical activity assessment, perceived stress, and wellbeing.                                                                                          -Thank you. This specific information has been added within the Methods section (lines 139–142; 154–156; 170; 173; 183–184).

In the section of the conclusion, I suggest the author should provide theoretical reflections and practical suggestions for the development of this pilot study.                                                          -Thank you. This additional content has now been added within the Conclusion section (lines 486–493).

Round 2

Reviewer 1 Report

Thank you for addressing my prior comments and suggestions. The quality and readability of the manuscript have certainly improved. The reviewer just noted two grammatical edits that should be made. 

Pg 4, lns 161-162: This sentence is not complete. It appears it is missing a word or words. 

Pg 4, ln 180: removed “been”

Author Response

Pg 4, lns 161-162: This sentence is not complete. It appears it is missing a word or words.                                                                                                                                            -Thank you. This sentence has been revised (lines 170–174).

Pg 4, ln 180: removed “been”                                                                                              -Thank you. This was removed (line 184).